# Cardiovascular risk and cognitive performance: A population-based cross-sectional study (NEDICES2-RISK)

Ester Tapias-Merino[1,2,3,4]*, María del Canto De Hoyos-Alonso[2,5],
Javier Rubio-Serrano[2,6], Beatriz Arregui-Gallego[6], Ileana Gefaell-Larrondo[2,6],
Emiliano Rodríguez-Sánchez[2,7,8], Israel Contador[9,10], Teresa Sanz-Cuesta[2,6,11],
Félix Bermejo-Pareja[12,13], Isabel del Cura-González[2,6,10,11,14], NEDICES2-RISK Group

**1** Deputy Management of Health Care, Primary Care Assistance Management, Madrid, Spain,
**2** Research Network on Chronicity, Primary Care and Health Promotion (RICAPPS), Instituto de Salud Carlos III (ISCIII), Madrid, España, **3** Group for Research in Health Services and Results, Research Institute Hospital 12 de Octubre i+12, Madrid, Spain, **4** Department of Medicine, Faculty of Medicine, Complutense University of Madrid, Madrid, Spain, **5** Healthcare Centre Pedro Laín Entralgo, Primary Care Assistance Management, Madrid Health Service, Madrid, Spain, **6** Research Unit, Primary Care Assistance Management, Madrid Health Service, Madrid, Spain, **7** Research Unit for Primary Health Care, Institute for Biomedical Research of Salamanca (IBSAL), Castilla y León Health Service, Salamanca, Spain, **8** Department of Medicine, University of Salamanca, Salamanca, Spain, **9** Department of Basic Psychology, Psychobiology and Methodology of Behavioral Sciences, Faculty of Psychology, University of Salamanca, Salamanca, Spain, **10** Ageing Research Center, Karolinska Institute, Solna, Sweeden, **11** Gregorio Marañón Health Research Institute, Madrid Health Service, Madrid, Spain, **12** Research Institute Hospital 12 de Octubre i+12, Neurosciences Group CIBERNED, Madrid, Spain, **13** Consulting Neurologist, Hospital 12 de Octubre, Madrid, Spain, **14** Department of Medical Specialities and Public Health, Rey Juan Carlos University, Madrid, Spain

¶ Membership of the NEDICES2-RISK group is provided in the Acknowledgements.
* ester.tapias@salud.madrid.org

## Abstract

### Background

Cardiovascular risk (CVR) factors increase the risk of cognitive impairment. Our objective was to assess the relationship between CVR and cognitive performance by sex in the Spanish population.

### Methods

Cross-sectional analysis nested within the population-based NEDICES-2 cohort. The included participants were 55–75 years old without previous cardiovascular events. CVR was estimated using the REGICOR and FRESCO risk equations. Outcome variable: performance on premorbid intelligence, global cognition, memory, verbal fluency, visuoconstruction, attention, and psychomotor speed. Generalized linear models were built.

**Data availability statement:** Data cannot be shared publicly because of the Research Ethics Committee of the 12 de Octubre University Hospital approved this research without considering the option of data sharing. The data contains sensitive clinical information about the patients, and therefore there are ethical and legal restrictions on sharing it. Data are available from the Foundation for Research and Innovation in Primary Care (FIIBAP) of the Community of Madrid at fiibap@salud.madrid.org for researchers who meet the criteria for access to confidential data.

**Funding:** This study was financially supported by the Instituto de Salud Carlos III (ISCIII), through project PI11/01508, and by the Instituto de Salud Carlos III (ISCIII), through project PI18/00522, and co-funded by the European Union in the form of awards received by ET-M. The project has been peer-reviewed by the funding agency. Its publication has been funded by the Instituto de Salud Carlos III (ISCIII) through the RD21/0016/0026 (RICAPPS) group and by the European Union through the European Recovery Instrument ('Next Generation EU'), in the framework of the Recovery, Transformation and Resilience Plan in the form of awards received by ET-M. This study was also financially supported by the Fundación para la Investigación e Innovación Biomédica de Atención Primaria (FIIBAP) (https://www.fiibap.org) in the Community of Madrid, via their call for grants in 2018 and 2019, received by ET-M. This study was also financially supported by the XLI Congress of the Spanish Society of Family and Community Medicine: PAPPS/semFYC- Fundación Mutual Médica (https://www.semfyc.es/secciones/investigacion/becas/premio-pappssemfycfundacion-mutual-medica) research award received by ET-M. The article processing charge was funded by Fundación para la Investigación e Innovación Biomédica de Atención Primaria (FIIBAP). The funders had no additional role in study design, data collection and analysis, decision to publish, or preparation of the manuscript.

**Competing interests:** The authors have declared that no competing interests exist.

## Results

863 participants (56.0% women). Women with higher CVR estimated with FRESCO showed worse cognitive outcomes on Immediate memory test OR: 4.77 (CI95% 1.81–13.80), Delayed memory test OR: 2.86 (CI95% 1.09–7.91), Clock drawing test OR 3.23 (CI95% 1.07–10.10), TMTA-1 timing OR: 6.92 (CI95% 2.43–20.90) and TMTA-2 timing OR 3.89 (CI95% 1.34–11.50%). With moderate versus low CVR they scored worse on Clock drawing test OR 2.08 (CI95% 1.08–4.02), TMTA-1 timing OR: 2.99 (CI95% 1.62–5.58) and TMTA-2 timing OR 2.01 (CI95% 1.06–3.83%). Men with high versus low CVR with REGICOR had lower scores on MMSE-37 OR: 2.41 (CI95% 1.08–5.36), Word accentuation OR: 2.90 (CI95% 1.28–6.67), TMTA-1 OR: 3.68 (CI95% 1.57–8.71) and TMTA-2 OR: 2.82 (CI95% 1.24–6.41). With moderate versus low CVR, they scored worse on TMTA-1 OR: 2.21 (CI95% 1.15–4.41). With FRESCO, men with high versus low CVR were worse on Immediate memory OR: 2.23 (CI95% 1.07–4.70), Word accentuation OR: 2.99 (CI95% 1.25–7.53), and TMTA-1 timing 3.72 (CI95% 1.46–10.3) and the men with moderate versus low CVR. performed worse on MMSE-37 OR: 2.99 (CI95% 1.19–8.27).

## Conclusions

Higher CVR was associated with worse cognitive performance in the Spanish population 55–75 years old. The most affected cognitive domains were memory, attention and psychomotor speed.

## Introduction

The aging of the population leads to a rise in chronic conditions, including neurodegenerative diseases such as dementia [1]. The incidence and prevalence rates of dementia in Spain are analogous to other European countries, albeit within the lower spectrum [2–4]. Since the beginning of the 21st century, the vast majority of cohort studies and reviews on dementia have shown a decline in its incidence in the most affluent European countries and the United States, but not in other countries such as Japan [5–8]. While the trend in prevalence follows a similar pattern, it is less definitive. In Spain, the existing evidence only points to a decrease in prevalence among men [9].

Cognitive impairment and dementia pose significant costs and a negative effect on society, which is exacerbated by the aging of the population. Any successful intervention that reduces the incidence or delays the onset of cognitive impairment would have a notable impact on society. The risk of developing dementia and cognitive impairment in the elderly increases in the presence of cardiovascular disease and CVR factors [10–12].

For this reason, efforts are intensifying to identify risk factors that contribute to the appearance of cognitive impairment and dementia with the aim of intervening in those that are potentially modifiable. Several studies indicate that controlling

modifiable risk factors could prevent 30–73% of dementia cases [13–15]. CVR factors and lifestyle habits amount to up to two-thirds of the modifiable factors for the prevention of Alzheimer's disease [16]. Although the pathophysiological mechanisms of dementia remain unclear [17], several studies in general population samples have shown that microvascular dysfunction (assessed through biomarkers, imaging of small vessel disease or arterial stiffness) is associated with poorer performance in specific domains such as memory and processing speed [18–22].

The prevalence of CVR factors in Spain is high [23]. However, a low coronary risk has been reported in the Spanish population [24,25] and the incidence of stroke is lower than in other countries [26–28]. Therefore, based on current figures of <100 deaths from cardiovascular disease per 100,000 inhabitants, Spain is classified as a low CVR country by the World Health Organization [29].

Different risk formulas analyzing cohort studies over a follow-up of >10 years have been developed to estimate CVR [30,31]. These CVR scores must be adapted to the characteristics and heterogeneity of different populations [32]. In Spain, two risk functions are validated to estimate the CVR: the Framingham REGICOR (Registre Gironí del Cor) tables for CVR [30] and the FRESCO (Función de Riesgo Española de acontecimientos Coronarios y Otros) equation [31]. The combined effect of a series of CVR factors on cognition has also been studied employing these risk foromulas, indicating that the higher the CVR, the lower the cognitive performance [33–40].

In this context, a significant portion of the existing scientific literature on cardiovascular medicine was conducted in samples predominantly composed of men. Certain studies have pointed to differences in CVR factors by sex [41,42] and gender [43,44]. Moreover, studies have reported a greater predisposition of women to develop cerebral small vessel disease [45,46] and the effect of CVR on cognition appears to differ between sexes [47]. However, the research conducted in the Spanish population using these tools is limited.

The main objective of the current study is to assess the relationship between CVR and cognitive performance by sex in a subsample of the NEDICES-2 cohort.

## Materials and methods

The study is a cross-sectional analysis nested within the population-based NEDICES-2 cohort.

### Study population

The study population was drawn from the NEDICES-2 (Neurological Disorders in Central Spain) cohort study, primarily designed to assess the main chronic neurological diseases. This cohort was composed of people aged ≥55 years who were assigned to the participating physicians in six health centers within the Spanish Health System spanning four provinces (Salamanca, Ávila, Segovia, and Madrid). Participants were selected by random sampling stratified by sex and age in 2011. Blood, urine, saliva, and hair samples were gathered for a biobank upon enrolment [48].

The present study included subjects 55–75 years of age without dementia who were part of the NEDICES-2 cohort and had undergone neuropsychological assessment. The resulting sample was comprised of 962 patients that constitute the NEDICES2-RISK cohort. Of these participants, 94 had suffered cardiovascular events (ischemic heart disease, stroke, and/or peripheral artery disease) and were therefore excluded from the analysis, as the CVR functions can only be utilized in the absence of such events [49]. Although participants were selected based on being between 55 and 75 years of age at the time of inclusion in the study, this age criterion referred specifically to their age at enrollment. Due to organizational and logistical aspects of the study implementation, there was an approximate two-year gap between participant inclusion and the cognitive assessment interviews. As a result, some participants had exceded the upper age limit by the time the REGICOR scores was analyzed, which explains the exclusion of 68 individuals aged 75 years or older at assessment with REGICOR. Following the FRESCO guidelines, 308 patients under treatment with lipid-lowering agents were also excluded from the sample whose CVR was estimated using this function [31]. Therefore, for the calculation of cardiovascular risk using the REGICOR score, a total of 795 patients were ultimately analyzed using REGICOR and 555 using FRESCO Fig 1 shows the flowchart of participants throughout the study.

## Sociodemographic and clinical variables

Sociodemographic variables (age, sex, educational level) and lifestyle habits (smoking, physical activity) were recorded along with the following clinical variables: obesity, diabetes mellitus (DM), atrial fibrillation (AF), hypercholesterolemia, arterial hypertension (HTN), body mass index (BMI), total and high density lipoprotein (HDL-c) cholesterol, systolic blood pressure (SBP), diastolic blood pressure (DBP), depression (diagnosed in the medical record or by a positive outcome on the Center for Epidemiologic Studies Depression scale) [50], lipid-lowering treatment, antihypertensive treatment, and modulating treatments of the central nervous system (CNS) (anxiolytics, neuroleptics, antidepressants, and antiepileptic drugs).

Clinical data, including blood pressure, weight and height, were obtained through direct measurements performed by trained clinical staff during study visits. Information on participants´ medical history and comorbidities was collected by the study physicians and verified using medical records. When necessary, missing clinical information was completed through review of the electronic medical records.

**Cardiovascular risk equations.** Two risk functions that are validated in Spain were employed to estimate the CVR: the Framingham REGICOR (Registre Gironí del Cor) tables for CVR [30] and the FRESCO (Función de Riesgo Española de acontecimientos Coronarios y Otros) equation [31].

The Framingham REGICOR [30] is a version of the Framingham-Wilson equation calibrated by the REGICOR group. This tool was developed based on the population from the Northwest of Spain and incorporates the following CVR factors: age, sex, HTN, smoking habit, total cholesterol, and HDL-c. This function accurately predicts coronary disease events over 10 years. According to this risk function and the distribution by events, CVR can be stratified into four groups: low (<5%), moderate (5%–9.9%), high (10%–14.9%), and very high (≥15%) risk.

The FRESCO (Función de Riesgo Española de acontecimientos Coronarios y Otros) [31] function was developed using data from 12 Spanish cohorts. It uses both classic and non-laboratory CVR factors to predict the occurrence of coronary heart disease, stroke, and global cardiovascular events over 10 years in a typical southern-European population aged 35–79 years. The FRESCO function classifies more people as high risk than REGICOR function [51]. This study employed the FRESCO model that evaluates age, smoking habit, DM, SBP, total cholesterol, HDL-c, interaction between hypertensive treatment and SBP > 120, interaction age-smoking, and interaction age-SBP. Patients undergoing treatment with lipid-lowering drugs are excluded from this function. The threshold figures that classify patients according to their CVR using the FRESCO function vary with sex as follows: low (<3.3), moderate (≥3.3 and ≤6.9), high (>6.9 and ≤10.8), and very high (>10.8) risk in the case of men; and low (<2), moderate (≥2 and ≤3.9), high (>3.9 and ≤5.8), and very high (>5.8) risk in the case of women.

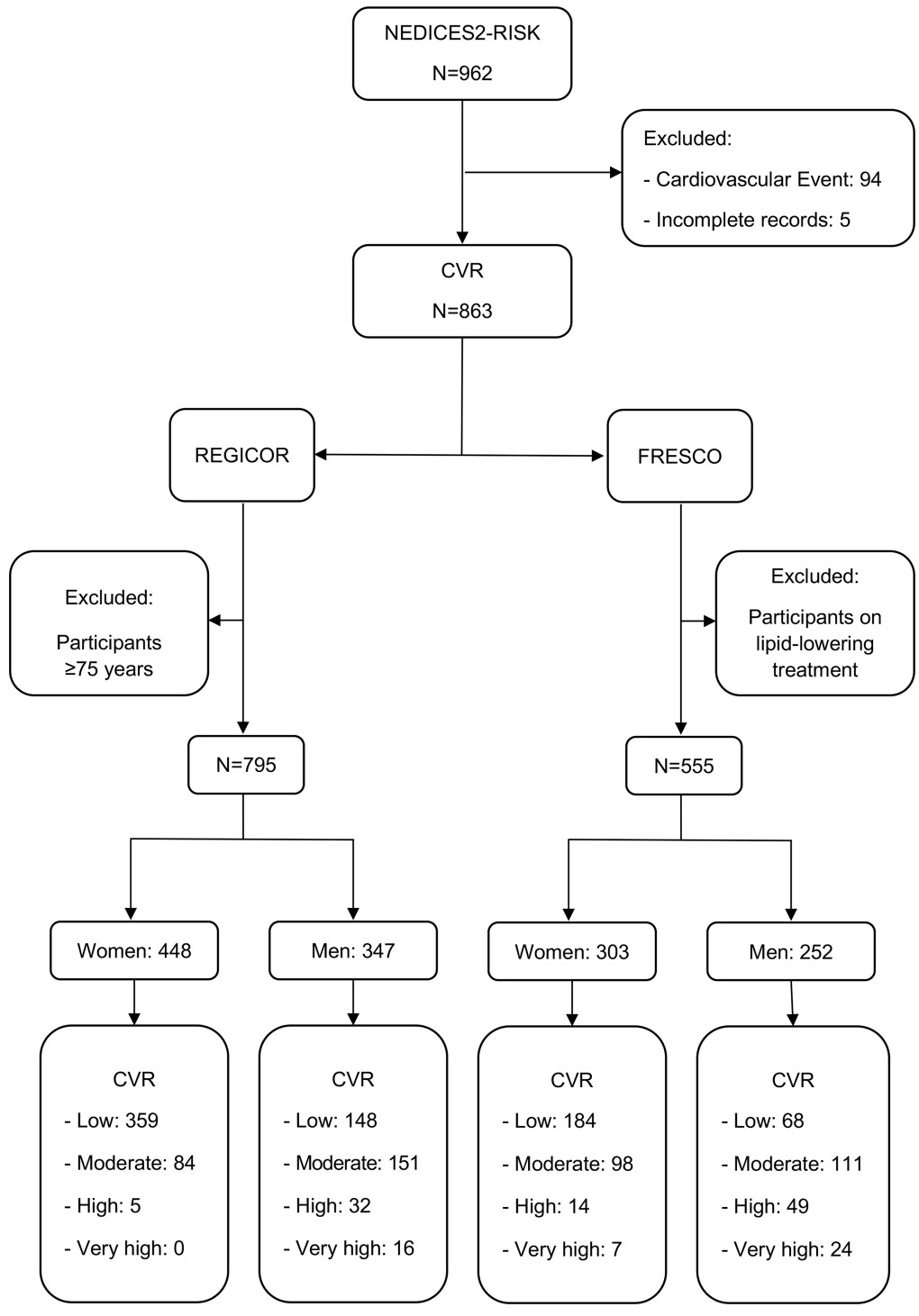

**Fig 1. Study flowchart of participants.** CVR: Cardiovascular risk.

**Neuropsychological tests.** A comprehensive and standardized test battery employed in the NEDICES-2 study [52] was used to explore different cognitive domains. These neuropsychological tests were administered in person during structured interviews conducted by qualified personnel.

**Global cognition** was explored using the Mini-Mental State Examination 37-item version (MMSE-37) [53–56], a modified and extended version that has been validated in the Spanish population. It contains 37 items that evaluate time and space orientation, memory, attention, calculation, language, object recognition, basic commands, and visuoconstructive capacity.

**Immediate and delayed memory** performance was determined with Six Object Memory Recall Test [57–59]. The subjects are shown six images depicting common objects to evaluate object recognition, immediate memory, and delayed recall after 5 minutes.

**Premorbid intelligence** through the Word Accentuation test [60], a Spanish adaptation of the Adult Reading Test [61]. This test assesses verbal intelligence by having the participant read 30 infrequent words without graphic accents and asking them to properly pronounce them.

**Verbal fluency**. A brief version was delivered to determine the number of elements in a specific category (animals) [62,63] that a subject can recall in one minute. Basically, this test examines semantic memory and verbal planning capacity.

**Visuoconstruction** was measured with the Clock Drawing test. This brief test explores comprehension, attention, visual and semantic memory, abstraction, planning and visuoconstructive praxis [64].

**Attention and psychomotor speed** using the Trail Making Test series A (TMTA) [65,66]. This visual-motor integration test provides information on visual search, scanning, processing speed, mental flexibility, and executive functions. In part 1, the subject has to connect in order the numbers from 1 to 25 that are randomly drawn on a paper. In part 2, numbers from 1 to 25 are drawn in duplicate, half in a white circle and half in a black circle, and the participant is asked to connect the first number in white with its black counterpart and then the next white number with its black counterpart until the 25 numbers are completed. Both the number of errors and execution time to complete the task are recorded.

In all the described tests, higher scores indicate better cognitive performance, except in the case of the TMTA, in which longer execution times or a greater number of errors imply worse cognitive performance.

## Statistical analysis

The sociodemographic, lifestyle, and clinical characteristics of subjects without cardiovascular events were described by their frequencies and percentages for qualitative variables and their median and interquartile range (IQR) for quantitative ones. Subsequently, the variables were compared disaggregated by sex, using the Chi-squared or Mann-Whitney U tests depending on the type of variable. The baseline characteristics and neuropsychological tests outcomes of patients excluded from the main analysis due to previous cardiovascular event were also analyzed.

The CVR results were categorized into low, moderate, high, and very high risk according to the REGICOR and FRESCO equations. Three categories of CVR were established for the analysis, with patients with high and very high CVR being clustered into a unique "high CVR" category. The baseline characteristics were compared between the three groups using the Chi-squared or Fisher's exact and Kruskal-Wallis or Mann-Whitney U tests, since the values of all the examined variables, including the neuropsychological test scores, did not fit a normal distribution. The same statistical methods were employed to compare the performance in the different neuropsychological tests and the cohort characteristics at the baseline stratified by CVR and cognitive performance.

CVR factors were described by comparing the lowest quartile of the sample (worst) in terms of cognitive performance versus the rest (≤P25 vs. >P25 in all tests except for the TMTA, where the quartile ≥P75 indicates the worst performance). The decision to divide cognitive scores into quartiles (≤25th percentile vs. >25th) was made because, at the start of the study, participants had very similar cognitive scores, resulting in a non-normal distribution and because quartile

stratification allows for more accurate identification of participants with lower cognitive performance compared to those with higher performance, facilitating comparisons between groups and offering results that are more easily interpretable from a clinical point of view.

All the outcomes from the analyses were disaggregated by sex.

A Directed Acyclic Graph (DAG) was constructed using DAGitty v3.1 to identify possible paths of confusion between exposure (cardiovascular risk) and outcome (cognitive performance) (S1 Fig). In addition, A generalized linear model using binomial distributions was constructed for each CVR function and each neuropsychological test to explore the influence of CVR on cognitive performance. Odds ratios (OR) were calculated with the corresponding confidence intervals (CI) set at 95%. The dependent variable was the score on each test, all of which were categorized as a dichotomous variable. This allowed comparing the lowest quartile (worst) of cognitive performance versus the rest (≤P25 vs. >P25 in all tests except for the TMTA test, where the quartile ≥P75 indicates the worst performance). The independent variable was the CVR, which was classified into low, moderate, and high as previously explained. Models were adjusted for the following confounding variables: educational level (no education or primary education vs. secondary or superior education), obesity (presence if BMI ≥ 30 or absence), AF (presence or absence), depression (presence or absence), physical exercise (active or inactive), and treatment modulating the CNS (anxiolytics, neuroleptics, antidepressants, and antiepileptic drugs). Some potential covariates were not included in the model, as they are integrated into the risk value calculation in the REGICOR and FRESCO tools, and their additional inclusion could induce collinearity.

All statistical analyses were performed using R-Studio [67] (version 4.3.1) software with the significance level established at $p < 0.05$.

## Ethics approval

This study was conducted following the principles of the Declaration of Helsinki and its subsequent revisions and received approval (February 12th, 2019) from the Ethics Committee for Research of the Hospital Universitario 12 de Octubre (ref. CEIm 19/021). It was also granted a favorable report by the Central Committee for Research of the Primary Care Management of Madrid (February 20th, 2019).

## Results

### Excluded patients

Of the 962 patients included in the NEDICES2-RISK cohort, 94 subjects who presented cardiovascular disease (ischemic heart disease, stroke, or peripheral artery disease) were not included in the study. Despite not being part of this research, a comparison was made of the sociodemographic and clinical characteristics and neuropsychological test results between these 94 patients (21 women and 73 men) and the remaining 868 (486 women and 382 men) who had not suffered any cardiovascular event (S1 Table). Both men and women with cardiovascular disease were older and less physically active and a greater proportion of them presented HTN and dyslipidemia. The comparison between men with and without cardiovascular disease showed that the former group presented higher proportions of smokers or ex-smokers (90.5% vs. 72.1%, $p = 0.002$), DM (43.8% vs. 22.0%, $p = 0.001$), and AF (15.1% vs 6.3%, $p = 0.019$). Both men and women with cardiovascular disease had lower DBP and total cholesterol figures, as well as HDL-c in the case of men (43 vs. 49 mg/dl, $p = 0.003$).

In terms of cognitive performance, women with cardiovascular disease performed worse on the Delayed Recall (median: 4 vs. 5 points, $p = 0.036$) and more errors in the TMTA-1 (median: 1 vs. 0, $p = 0.002$) and part 2 (median: 1 vs. 0, $p = 0.011$). In contrast, no significant differences were found between men with or without cardiovascular disease in the results of any of the neuropsychological tests.

## Characteristics of the sample

The NEDICES2-RISK cohort included a total of 863 participants who had not experienced previous cardiovascular events, 483 (56.0%) women and 380 (44.0%) men. The median age of the overall sample was 67 years (IQR: 62.00–71.00). The level of education was lower in women, with 65.9% having primary education or less compared to 57.6% of men. In terms of lifestyle habits, 66.5% of women were not physically active versus 62% of men and the percentage of smokers and ex-smokers was 72.3% in men compared to only 34.4% in women. Regarding clinical variables, a higher percentage of men than women suffered DM (22.1% vs. 12.8%), HTN (46.1% vs. 45.3%), and AF (6.3% vs. 2.5%). Men also exhibited higher figures of BMI (28.6 vs. 27.6), SBP (132 vs. 130 mmHg), and DBP (78 vs. 75 mmHg) than women. Conversely, a greater percentage of women had depression (18.0% vs. 8.7%) or were prescribed with CNS treatments (30.0% vs. 16.6%). Women were also more likely to present dyslipidemia (50.3% vs. 45.0%) and higher total cholesterol levels (208.0 vs. 192.0 mg/dl) but showed better HDL-c figures (57 vs. 49 mg/dl) (Table 1).

## Cardiovascular risk

High or very high CVR was present in 48 (13.8%) men and 5 (1.1%) women according to the REGICOR function and in 73 (29.0%) men and 21 (6.9%) women according to the FRESCO equation (Table 1). Tables 2 and 3 show the relationships between the examined variables and CVR. In addition to the observed relationships between higher CVR and the variables used for its calculation through the FRESCO and REGICOR functions, other specific significant differences were found with higher BMI in women, lower level of education in men and sedentary lifestyle using the FRESCO equation.

## Cognitive performance

When analyzing the scores of each neuropsychological test in relation to the sociodemographic and clinical variables of the patients (S2-S11 Tables), all cognitive outcomes worsened with increasing age for both men and women. They also worsened with less schooling, except for the Immediate and Delayed Recall tests, for women. In women, a sedentary lifestyle and HTN negatively affected the outcomes on the MMSE-37, Word Accentuation test, Verbal Fluency, Clock Drawing test, and TMTA completion times and errors; diabetes negatively impacted the scores of the MMSE-37, Word Accentuation, verbal fluency, and TMTA timing tests; higher BMI lowered the scores on the Word Accentuation, Verbal Fluency, and TMTA-2 timing tests; and worse HDL-c entailed worse performance on the MMSE-37, Word Accentuation, Clock Drawing, and TMTA timing tests. Sedentary men scored worse on the MMSE-37 and had more errors on the TMTA-1 while men with diabetes performed worse on the Verbal Fluency test and TMTA-2 timing. Smokers and ex-smokers of both sexes obtained better results in most of the tests, with significant differences in the female sample in the MMSE-37, Word Accentuation, Verbal Fluency, and Clock Drawing tests and for both women and men in the TMTA timing. No effect of AF was observed on any test outcome.

In terms of non-cardiovascular variables, women undergoing treatment with CNS modulating drugs obtained lower scores in the delayed memory test and more errors in the TMTA parts 1 and 2. Men with depression had worse results in the Clock Drawing test while those with prescribed CNS treatment had more errors in the TMTA-1.

## Cognitive performance by cardiovascular risk

In women (Table 4), no relationship was observed between the CVR calculated with REGICOR and scores of the neuropsychological tests. When calculating CVR with the FRESCO equation, cognitive performance was lower in women with high CVR compared to those with low and moderate CVR according to the Global cognition (MMSE–37) (32 vs 34 and 32.5 points, $p = 0.005$), the Immediate Recall test (4 vs. 5 points, $p = 0.004$), Word Accentuation test (21 vs. 26 and 24 points, $p = 0.044$), Clock Drawing test (3 vs. 4 points, $p = 0.001$), TMTA-1 timing (83 vs. 55 and 68 s, $p < 0.001$), and TMTA-2 timing (96 vs. 62 and 74 s, $p = 0.009$).

**Table 1. Sociodemographic and clinical characteristics by sex.**

| Variables | Overall (N = 863) | Women (n = 483) | Men (n = 380) | p |
|---|---|---|---|---|
| **Sociodemographic** | | | | |
| **Age[1]** | 67.0 [62.0-71.0] | 67.0 [62.0-71.0] | 67.0 [62.0-71.0] | 0.515[a] |
| **Education level[2]** | | | | |
| No education-Primary | 531 (62.3) | 315 (65.9) | 216 (57.6) | 0.016[b] |
| Secondary-Superior | 322 (37.7) | 163 (34.1) | 159 (42.4) | |
| **Lifestyle** | | | | |
| **Sedentary lifestyle[2]** | 552 (64.5) | 319 (66.5) | 233 (62.0) | 0.197[b] |
| **Smoking[2]** | | | | |
| Non-smoker | 420 (49.0) | 315 (65.6) | 105 (27.8) | <0.001[b] |
| Smoker | 114 (13.3) | 59 (12.3) | 55 (14.6) | |
| Ex-smoker | 324 (37.8) | 106 (22.1) | 218 (57.7) | |
| **Chronic diseases** | | | | |
| **Hypertension[2]** | 394 (45.7) | 219 (45.3) | 175 (46.1) | 0.889[b] |
| **Diabetes mellitus[2]** | 146 (16.9) | 62 (12.8) | 84 (22.1) | <0.001[b] |
| **Dyslipidemia[2]** | 414 (48.0) | 243 (50.3) | 171 (45.0) | 0.138[b] |
| **Atrial fibrillation[2]** | 36 (4.2) | 12 (2.5) | 24 (6.3) | 0.009[b] |
| **Depression[2]** | 120 (13.9) | 87 (18.0) | 33 (8.7) | <0.001[b] |
| **CNS treatment[2]** | 208 (24.1) | 145 (30.0) | 63 (16.6) | <0.001[b] |
| **Physical examination** | | | | |
| **BMI[1]** | 28.1 [25.5-30.6] | 27.6 [24.9-30.6] | 28.6 [26.7-30.7] | <0.001[a] |
| **SBP[1]** | 130.0 [120.0-140.0] | 130.0 [120.0-140.0] | 132.0 [121.0-140.0] | 0.007[a] |
| **DBP[1]** | 77.0 [70.0-81.0] | 75.0 [70.0-80.0] | 78.0 [70.0-85.0] | 0.010[a] |
| **Total cholesterol[1]** | 200.0 [175.5-225.0] | 208.0 [183.5-231.0] | 192.0 [167.0-215.0] | <0.001[a] |
| **HDL–c[1]** | 53.0 [45.0-63.0] | 57.0 [49.0-67.0] | 49.0 [40.0-57.0] | <0.001[a] |
| **Cardiovascular risk** | | | | |
| **REGICOR[2]** | | | | |
| Low | 507 (63.8) | 359 (80.1) | 148 (42.7) | <0.001[b] |
| Moderate | 235 (29.6) | 84 (18.8) | 151 (43.5) | |
| High | 53 (6.7) | 5 (1.1) | 48 (13.8) | |
| **FRESCO[2]** | | | | |
| Low | 252 (45.4) | 184 (60.7) | 68 (27.0) | <0.001[b] |
| Moderate | 209 (37.7) | 98 (32.3) | 111 (44.0) | |
| High | 94 (16.9) | 21 (6.9) | 73 (29.0) | |

CVR: Cardiovascular risk; CNS treatment: treatments that modulate the central nervous system; BMI: Body mass index; SBP: Systolic blood pressure (mmHg); DBP: Diastolic blood pressure (mmHg); HDL-c: High Density Lipoprotein cholesterol; REGICOR (Registre GIroní del COR); FRESCO: Spanish Coronary events Function; 1: median [Q1–Q3]; 2: n (%); a: Mann-Whitney U test; b: Chi–squared test.

In men (Table 4), using the REGICOR function for estimating CVR, neurological test scores progressively worsened with high CVR compared to those with low and moderate CVR in the MMSE-37 (33 vs. 34.5 and 35, $p = 0.006$), Word Accentuation test (23 vs. 26 and 25, $p < 0.001$), and execution times in the TMTA part 1 (66s vs. 48s and 58s, $p < 0.001$) and part 2 (76s vs 57s and 64s, $p = 0.002$). When CVR was determined with the FRESCO equation, similar results were found. Those at high cardiovascular risk performed worse than those at low risk in the MMSE-37 (33 vs. 35, $p = 0.004$), delayed recall memory (4 vs. 5, $p = 0.036$), Word Accentuation test (22.5 vs. 26 and 25, $p = 0.002$), verbal fluency test (17 vs. 20 and 19, $p = 0.03$), TMTA-1 timing (70.5s vs. 49.5s and 55s, $p < 0.001$) and TMTA-2 timing (73s vs 56s and 62s, $p = 0.004$).

**Table 2. Sociodemographic and clinic characteristics in women by cardiovascular risk.**

| Variables | REGICOR | | | | FRESCO | | | |
|---|---|---|---|---|---|---|---|---|
| | Low CVR (n = 359) | Moderate CVR (n = 84) | High CVR (n = 5) | *p* | Low CVR (n = 184) | Moderate CVR (n = 98) | High CVR (n = 21) | *p* |
| Age[1] | 67.0 [61.0-71.0] | 65.0 [62.0-68.0] | 66.0 [66.0-69.0] | 0.420[a] | 64.0 [60.0-69.0] | 69.0 [65.0-73.0] | 71.0 [68.0-75.0] | <0.001[a] |
| Education level[2] | | | | | | | | |
| No education-Primary | 231 (64.9) | 53 (64.6) | 4 (80.0) | 0.867[b] | 109 (59.9) | 71 (72.4) | 15 (71.4) | 0.089[c] |
| Secondary-Superior | 125 (35.1) | 29 (35.4) | 1 (20.0) | | 73 (40.1) | 27 (27.6) | 6 (28.6) | |
| Sedentary lifestyle[2] | 231 (64.7) | 56 (67.5) | 3 (60.0) | 0.867[b] | 112 (61.5) | 73 (74.5) | 18 (85.7) | 0.017[b] |
| Smoking[2] | | | | | | | | |
| Non-smoker | 235 (65.8) | 45 (54.2) | 3 (60.0) | <0.001[b] | 120 (65.6) | 75 (76.5) | 12 (57.1) | <0.001[b] |
| Smoker | 35 (9.8) | 23 (27.7) | 1 (20.0) | | 13 (7.1) | 15 (15.3) | 8 (38.1) | |
| Ex-smoker | 87 (24.4) | 15 (18.1) | 1 (20.0) | | 50 (27.3) | 8 (8.2) | 1 (4.8) | |
| Hypertension[2] | 143 (39.8) | 54 (64.3) | 5 (100) | <0.001[b] | 54 (29.3) | 52 (53.1) | 15 (71.4) | <0.001[c] |
| Diabetes mellitus[2] | 18 (5.0) | 32 (38.1) | 5 (100) | <0.001[b] | 2 (1.1) | 7 (7.1) | 9 (42.9) | <0.001[b] |
| Dyslipidemia[2] | 172 (47.9) | 49 (58.3) | 4 (80.0) | 0.114[b] | 37 (20.1) | 20 (20.4) | 6 (28.6) | 0.659[c] |
| Atrial fibrillation[2] | 6 (1.7) | 4 (4.8) | 0 (0) | 0.197[b] | 2 (1.1) | 3 (3.1) | 0 (0) | 0.544[b] |
| Depression[2] | 64 (17.8) | 17 (20.2) | 1 (20.0) | 0.763[b] | 30 (16.3) | 15 (15.3) | 5 (23.8) | 0.631[c] |
| CNS treatment[2] | 100 (27.9) | 29 (34.5) | 1 (20.0) | 0.462[b] | 56 (30.4) | 25 (25.5) | 9 (42.9) | 0.271[c] |
| BMI[1] | 26.9 [24.5-30.3] | 29.1 [26.0-32.6] | 32.6 [32.4-34.0] | <0.001[a] | 26.8 [24.3-30.4] | 27.5 [25.4-29.6] | 32.4 [27.9-34.4] | 0.002[a] |
| SBP[1] | 129.0 [120.0-138.0] | 140.0 [128.0-145.0] | 145.0 [140.0-160.0] | <0.001[a] | 125.5 [116.8-139.3] | 130.0 [120.3-141.5] | 130.0 [120.0-140.0] | 0.036[a] |
| DBP[1] | 75.0 [70.0-80.0] | 80.0 [70.8-82.3] | 76.0 [70.0-80.0] | 0.019[a] | 75.5 [70.0-80.0] | 76.5 [71.3-80.0] | 76.0 [70.0-80.0] | 0.755[a] |
| Total cholesterol[1] | 210.0 [187.5-232.0] | 205.0 [172.3-228.5] | 181.0 [176.0-214.0] | 0.083[a] | 212.0 [193.8-233.0] | 218.0 [199.3-234.3] | 211.0 [183.0-237.0] | 0.586[a] |
| HDL–c[1] | 60.0 [53.0-69.0] | 46.0 [41.8-50.0] | 37.0 [37.0-42.0] | <0.001[a] | 64.0 [54.0-74.0] | 50.5 [44.0-59.0] | 48.0 [41.0-52.0] | <0.001[a] |

CVR: Cardiovascular risk; CNS treatment: treatments that modulate the central nervous system; BMI: Body mass index; SBP: Systolic blood pressure (mmHg); DBP: Diastolic blood pressure (mmHg); HDL-c: High Density Lipoprotein cholesterol; REGICOR (Registre GIroní del COR); FRESCO: Spanish Coronary events Function; 1: median [Q1–Q3]; 2: n (%); a: Kruskal-Wallis test; b: Fisher's test; c: Chi–squared test.

Fig 2 shows the relationship between cognitive performance measured by neuropsychological tests and CVR calculated with the REGICOR and FRESCO equations adjusted for variables with a potential effect on cognitive performance (educational level, sedentary lifestyle, obesity, AF, depression, and CNS-modulating treatment). The results for the non-adjusted and adjusted model are shown in S12 and S13 Tables.

Both men and women with high or moderate CVR showed worse cognitive performance compared to those with low CVR. In women (Fig 2), when CVR was estimated with the FRESCO equation, worse cognitive performance was observed in women with high CVR compared to those with low CVR as indicated by the scores in Immediate recall OR: 4.77 (CI95% 1.81–13.80), Delayed Recall OR: 2.86 (CI95% 1.09–7.91), clock drawing test OR 3.23 (CI95% 1.07–10.10), TMTA-1 timing OR: 6.92 (CI95% 2.43–20.90) and TMTA-2 timing OR 3.89 (CI95% 1.34–11.50%). The performance was also worse in women with moderate CVR versus those with low CVR by the scores in Clock drawing test OR 2.08 (CI95% 1.08–4.02), TMTA-1 timing OR: 2.99 (CI95% 1.62–5.58) and TMTA-2 timing OR 2.01 (CI95% 1.06–3.83%).

**Table 3. Sociodemographic and clinic characteristics in men by cardiovascular risk.**

| Variables | REGICOR | | | | FRESCO | | | |
|---|---|---|---|---|---|---|---|---|
| | Low CVR (n = 148) | Moderate CVR (n = 151) | High CVR (n = 48) | p | Low CVR (n = 68) | Moderate CVR (n = 111) | High CVR (n = 73) | p |
| Age[1] | 64.0 [60.0-68.0] | 67.0 [63.0-71.0] | 68.0 [64.0-70.3] | <0.001[a] | 62.0 [59.0-65.0] | 65.0 [60.5-70.0] | 70.0 [68.0-74.0] | <0.001[a] |
| Education level[2] | | | | | | | | |
| No education-Primary | 69 (47.6) | 91 (60.7) | 30 (63.8) | 0.037[b] | 34 (50.7) | 66 (60.0) | 52 (72.2) | 0.033[b] |
| Secondary-Superior | 76 (52.4) | 59 (39.3) | 17 (36.2) | | 33 (49.3) | 44 (40.0) | 20 (27.8) | |
| Sedentary lifestyle[2] | 83 (56.8) | 90 (59.6) | 33 (70.2) | 0.266[b] | 37 (55.2) | 63 (56.8) | 53 (74.6) | 0.025[b] |
| Smoking[2] | | | | | | | | |
| Non-smoker | 44 (29.9) | 44 (29.1) | 8 (17.0) | <0.001[b] | 19 (27.9) | 32 (28.8) | 25 (34.7) | 0.047[c] |
| Smoker | 8 (5.4) | 19 (12.6) | 26 (55.3) | | 4 (5.9) | 17 (15.3) | 15 (20.8) | |
| Ex-smoker | 95 (64.6) | 88 (58.3) | 13 (27.7) | | 45 (66.2) | 62 (55.9) | 32 (44.4) | |
| Hypertension[2] | 56 (37.8) | 69 (45.7) | 31 (64.6) | 0.005[b] | 9 (13.2) | 39 (35.1) | 57 (78.1) | <0.001[b] |
| Diabetes mellitus[2] | 13 (8.8) | 39 (25.8) | 25 (52.1) | <0.001[b] | 3 (4.4) | 9 (8.1) | 27 (37.0) | <0.001[c] |
| Dyslipidemia[2] | 62 (41.9) | 66 (43.7) | 26 (54.2) | 0.323[b] | 7 (10.3) | 19 (17.1) | 18 (24.7) | 0.080[b] |
| Atrial fibrillation[2] | 11 (7.4) | 9 (6.0) | 1 (2.1) | 0.453[c] | 1 (1.5) | 6 (5.4) | 4 (5.5) | 0.455[c] |
| Depression[2] | 11 (7.4) | 14 (9.3) | 5 (10.4) | 0.763[b] | 7 (10.3) | 7 (6.3) | 10 (13.7) | 0.240[b] |
| CNS treatment[2] | 25 (16.9) | 28 (18.5) | 6 (12.5) | 0.624[b] | 13 (19.1) | 15 (13.5) | 16 (21.9) | 0.311[b] |
| BMI[1] | 28.5 [26.6-30.4] | 28.7 [26.8-30.9] | 28.6 [27.0-30.8] | 0.745[a] | 28.4 [26.2-31.4] | 29.4 [26.5-31.0] | 28.4 [27.3-29.9] | 0.414[a] |
| SBP[1] | 125.5 [119.8-135.0] | 135.0 [125.0-142.0] | 143.0 [138.5-154.3] | <0.001[a] | 130.0 [120.0-138.5] | 130.0 [124.5-140.0] | 136.0 [127.0-145.0] | 0.006[a] |
| DBP[1] | 75.5 [70.0-80.0] | 80.0 [72.0-85.0] | 80.0 [75.0-90.0] | 0.002[a] | 76.0 [71.0-85.0] | 80.0 [70.0-85.0] | 76.0 [70.0-82.0] | 0.540[a] |
| Total cholesterol[1] | 186.0 [156.8-210.0] | 197.0 [172.5-222.0] | 210.0 [174.8-226.5] | <0.001[a] | 186.0 [164.5-214.3] | 198.0 [179.5-222.0] | 204.0 [176.0-218.0] | 0.125[a] |
| HDL–c[1] | 55.0 [47.0-65.0] | 44.0 [38.0-53.0] | 42.5 [36.0-48.0] | <0.001[a] | 55.0 [47.8-65.0] | 49.0 [40.0-56.0] | 44.0 [38.0-49.0] | <0.001[a] |

CVR: Cardiovascular risk; CNS treatment: treatments that modulate the central nervous system; BMI: Body mass index; SBP: Systolic blood pressure (mmHg); DBP: Diastolic blood pressure (mmHg); HDL-c: High Density Lipoprotein cholesterol; REGICOR (Registre GIroní del COR); FRESCO: Spanish Coronary events Function; 1: median [Q1–Q3]; 2: n (%); a: Kruskal-Wallis test; b: Chi–squared test; c: Fisher's test.

In men (Fig 2), when comparing the test results of those with high CVR against those with low CVR as calculated with the REGICOR tables, worse performance was observed in the MMSE-37 OR: 2.41 (CI95% 1.08–5.36), Word accentuation OR: 2.90 (CI95% 1.28–6.67), TMTA-1 OR: 3.68 (CI95% 1.57–8.71) and TMTA-2 OR: 2.82 (CI95% 1.24–6.41. The performance was also worse in men with moderate CVR versus those with low CVR in the TMTA-1 OR: 2.21 (CI95% 1.15–4.41). After comparing test scores of men with high and low CVR as calculated with the FRESCO equation, differences were observed in outcomes on Immediate Recall OR: 2.23 (CI95% 1.07–4.70), Word accentuation OR: 2.99 (CI95% 1.25–7.53), and TMTA-1 timing 3.72 (CI95% 1.46–10.3). Comparing test scores of men with moderate and low CVR, differences were observed in outcomes on MMSE-37 OR: 2.99 (CI95% 1.19–8.27).

Table 4. Scores on cognitive performance tests by cardiovascular risk and sex.

| | Global cognition (MMSE–37)[1] | Memory | | Premorbid intelligence (Word Accentuation)[1] | Verbal fluency (Verbal Fluency)[1] | Visuoconstruction (Clock Drawing)[1] | Attention and psychomotor speed | | | |
|---|---|---|---|---|---|---|---|---|---|---|
| | | Immediate Recall[1] | Delayed Recall[1] | | | | TMTA–1[1] | TMTA–2[1] | TMTA–Errors 1[1] | TMTA–Errors 2[1] |
| **Women** | | | | | | | | | | |
| **REGICOR** | | | | | | | | | | |
| Low CVR (n=359) | 33.0 [31.0–35.0] | 5.0 [4.0–6.0] | 5.0 [4.0–6.0] | 25.0 [22.0–28.0] | 17.0 [14.0–21.3] | 4.0 [3.0–4.0] | 59.0 [45.0–78.0] | 67.0 [50.0–99.0] | 0.0 [0.0–1.0] | 0.0 [0.0–2.0] |
| Moderate CVR (n=84) | 33.5 [30.0–36.0] | 5.0 [4.0–6.0] | 5.0 [4.0–6.0] | 25.0 [19.5–27.5] | 17.0 [14.0–21.0] | 4.0 [3.0–4.0] | 58.0 [43.5–77.8] | 69.5 [45.0–105.5] | 0.0 [0.0–0.5] | 0.0 [0.0–1.0] |
| High CVR (n=5) | 32.0 [30.0–33.0] | 5.0 [5.0–5.0] | 6.0 [5.0–6.0] | 23.0 [18.0–26.0] | 14.0 [12.0–18.0] | 4.0 [4.0–4.0] | 101.0 [58.0–150.0] | 112.0 [59.0–185.0] | 0.0 [0.0–0.0] | 1.0 [0.0–2.0] |
| Overall (N=448) | 33.0 [30.0–35.0] | 5.0 [4.0–6.0] | 5.0 [4.0–6.0] | 25.0 [21.0–27.3] | 17.0 [14.0–21.0] | 4.0 [3.0–4.0] | 58.5 [45.0–78.3] | 67.0 [50.0–100.3] | 0.0 [0.0–1.0] | 0.0 [0.0–2.0] |
| *p*[a] | 0.512 | 0.742 | 0.422 | 0.242 | 0.632 | 0.157 | 0.122 | 0.323 | 0.604 | 0.269 |
| **FRESCO** | | | | | | | | | | |
| Low CVR (n=184) | 34.0 [31.0–35.3] | 5.0 [4.0–6.0] | 5.0 [4.0–6.0] | 26.0 [22.0–28.0] | 17.0 [13.0–22.0] | 4.0 [4.0–4.0] | 55.0 [43.0–71.0] | 62.0 [47.5–87.0] | 0.0 [0.0–0.0] | 0.0 [0.0–2.0] |
| Moderate CVR (n=98) | 32.5 [29.0–35.0] | 5.0 [4.0–6.0] | 5.0 [4.0–6.0] | 24.0 [21.0–27.0] | 16.0 [14.0–21.0] | 4.0 [3.0–4.0] | 68.0 [46.0–92.0] | 74.0 [49.0–112.0] | 0.0 [0.0–1.0] | 1.0 [0.0–2.0] |
| High CVR (n=21) | 32.0 [28.0–36.0] | 4.0 [4.0–5.0] | 4.0 [4.0–6.0] | 21.0 [19.0–26.0] | 15.0 [12.0–19.0] | 3.0 [2.5–4.0] | 83.0 [73.0–127.0] | 96.0 [66.0–156.0] | 0.0 [0.0–2.0] | 1.0 [0.0–1.0] |
| Overall (N=303) | 33.0 [30.0–35.0] | 5.0 [4.0–6.0] | 5.0 [4.0–6.0] | 25.0 [21.0–28.0] | 17.0 [13.0–21.0] | 4.0 [3.0–4.0] | 59.0 [44.0–80.0] | 66.0 [48.0–99.0] | 0.0 [0.0–1.0] | 0.0 [0.0–2.0] |
| *p*[a] | 0.005 | 0.004 | 0.203 | 0.044 | 0.439 | 0.001 | <0.001 | 0.009 | 0.147 | 0.972 |
| **Men** | | | | | | | | | | |
| **REGICOR** | | | | | | | | | | |
| Low CVR (n=148) | 34.5 [33.0–36.0] | 5.0 [4.0–5.0] | 5.0 [4.0–5.3] | 26.0 [23.0–28.0] | 19.0 [15.0–22.3] | 4.0 [3.0–4.0] | 48.0 [35.0–65.5] | 57.0 [37.5–83.5] | 0.0 [0.0–0.0] | 0.0 [0.0–1.0] |
| Moderate CVR (n=151) | 35.0 [32.0–36.0] | 5.0 [4.0–6.0] | 5.0 [4.0–6.0] | 25.0 [21.0–28.0] | 18.0 [14.0–23.0] | 4.0 [3.0–4.0] | 58.0 [42.0–80.5] | 64.0 [43.0–102.5] | 0.0 [0.0–0.0] | 0.0 [0.0–1.0] |
| High CVR (n=48) | 33.0 [29.0–35.0] | 5.0 [4.0–6.0] | 5.0 [4.0–6.0] | 23.0 [18.0–26.3] | 17.0 [13.8–20.0] | 4.0 [2.0–4.0] | 66.0 [54.3–97.8] | 76.0 [50.0–117.8] | 0.0 [0.0–1.0] | 1.0 [0.0–2.0] |
| Overall (N=347) | 34.0 [32.0–36.0] | 5.0 [4.0–6.0] | 5.0 [4.0–6.0] | 26.0 [21.0–28.0] | 18.0 [15.0–23.0] | 4.0 [3.0–4.0] | 55.0 [40.0–74.8] | 62.0 [43.0–95.8] | 0.0 [0.0–0.0] | 0.0 [0.0–1.0] |
| *p*[a] | 0.006 | 0.111 | 0.518 | <0.001 | 0.084 | 0.254 | <0.001 | 0.002 | 0.132 | 0.184 |
| **FRESCO** | | | | | | | | | | |
| Low CVR (n=68) | 35.0 [33.0–36.0] | 5.0 [4.0–5.0] | 5.0 [4.0–6.0] | 26.0 [23.0–28.0] | 20.0 [16.0–23.3] | 4.0 [3.0–4.0] | 49.5 [40.8–66.5] | 56.0 [37.8–88.5] | 0.0 [0.0–0.0] | 0.0 [0.0–1.0] |
| Moderate CVR (n=111) | 35.0 [31.0–36.0] | 5.0 [4.0–6.0] | 5.0 [4.0–5.0] | 25.0 [21.0–27.8] | 19.0 [15.0–23.0] | 4.0 [3.0–4.0] | 55.0 [41.3–78.8] | 62.0 [41.3–102.3] | 0.0 [0.0–0.0] | 0.0 [0.0–1.0] |
| High CVR (n=73) | 33.0 [30.0–35.0] | 4.0 [4.0–5.0] | 4.0 [3.0–5.0] | 22.5 [19.0–27.0] | 17.0 [14.0–20.0] | 4.0 [3.0–4.0] | 70.5 [50.5–104.8] | 73.0 [51.0–126.0] | 0.0 [0.0–1.0] | 1.0 [0.0–2.0] |
| Overall (N=252) | 34.0 [31.0–36.0] | 5.0 [4.0–6.0] | 5.0 [4.0–5.0] | 25.0 [20.8–28.0] | 19.0 [15.0–23.0] | 4.0 [3.0–4.0] | 57.5 [42.0–79.3] | 62.0 [43.0–103.0] | 0.0 [0.0–0.0] | 0.0 [0.0–1.0] |
| *p*[a] | 0.004 | 0.207 | 0.036 | 0.002 | 0.030 | 0.100 | <0.001 | 0.004 | 0.305 | 0.209 |

CVR: Cardiovascular risk; MMSE 37: Minimental State Examination 37-item version; TMTA: Trail Making Test series A (seconds); 1: median [Q1–Q3]; REGICOR (Registre GIroní del COR); FRESCO: Spanish Coronary events Function; a: Kruskal-Wallis test.

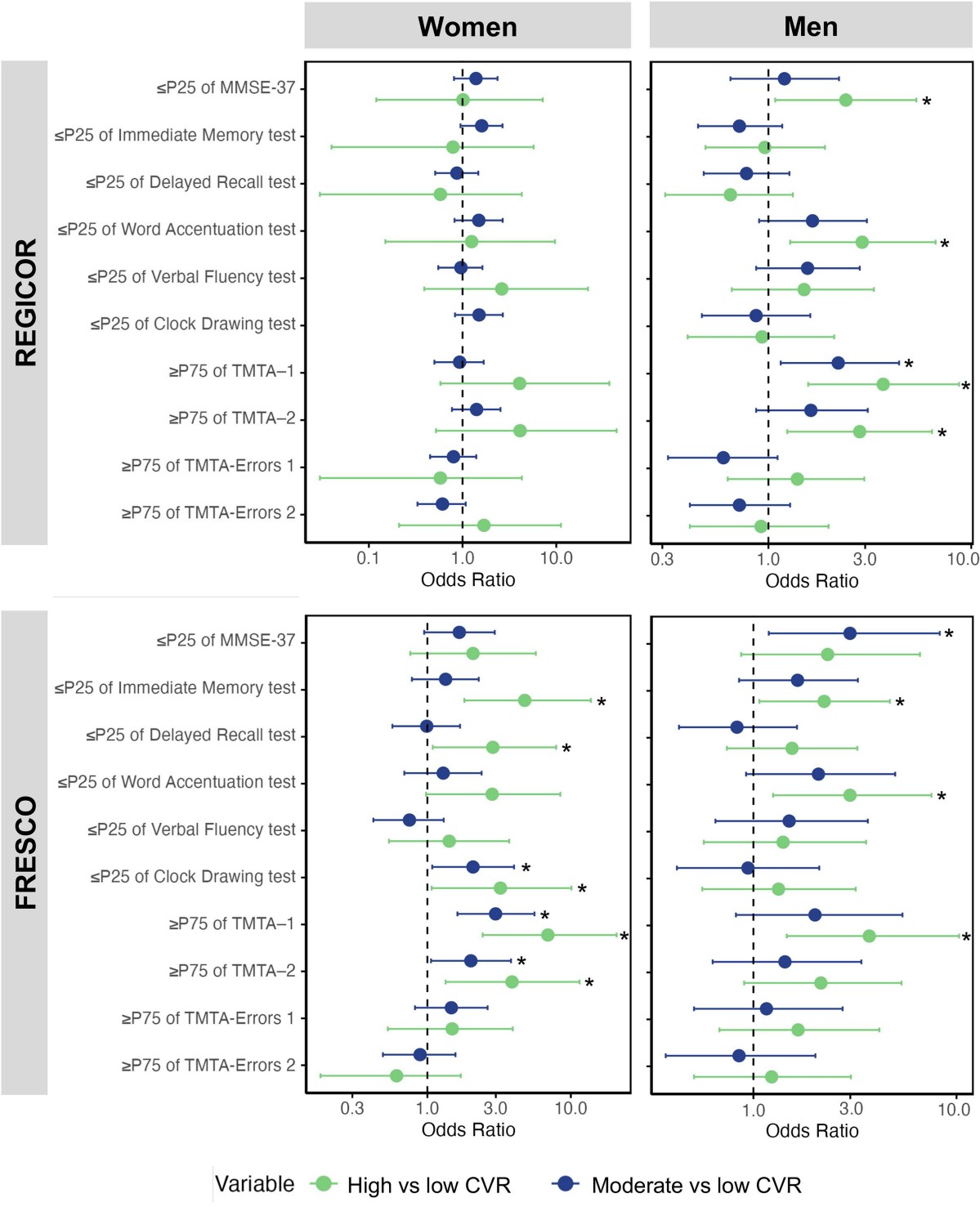

**Fig 2. Effect of cardiovascular risk measured with the REGICOR and FRESCO equations on cognitive performance disaggregated by sex.**
CVR: Cardiovascular risk; MMSE-37: Mini Mental State Examination-37; TMTA: Trail Making Test series **A.**

## Discussion

Dementia arises from the interaction between vascular dysfunction and neurodegenerative processes. Cerebrovascular dysfunction, especially in the microvasculature, causes cerebral hypoperfusion, impaired blood flow autoregulation, and structural damage to arterioles and capillaries. This generates oxidative stress, hypoxia, and mitochondrial dysfunction, affecting neuronal energy metabolism and causing synaptic damage.

The blood-brain barrier becomes permeable, allowing the filtration of toxic molecules and activating neuroinflammation through microglia and astrocytes. These vascular processes not only induce direct neuronal damage, but also enhance the classic mechanisms of dementia such as the accumulation of beta-amyloid and tau. Together, the interaction between vascular alterations, oxidative stress, neuroinflammation, and protein alterations leads to the progressive cognitive decline characteristic of dementia [68].

Our findings linking cardiovascular risk with cognitive performance are consistent with the growing body of evidence supporting the role of vascular pathophysiology in cognitive decline and dementia.

### Main findings

The present study found that moderate or high CVR, as determined using the REGICOR and FRESCO risk equations, correlated with poorer cognitive performance in the majority of the cognitive tests analyzed in both men and women.

### Comparison with other studies

In this cohort, women exhibited lower educational level, more sedentary lifestyles, more hypercholesterolemia, higher total cholesterol levels, increased depression rates, and were prescribed more CNS treatments than men. In contrast, the proportions of men were higher for smokers and ex-smokers (twice than in women), DM, HTN, and AF, and presented higher values of BMI, SBP, and DBP and lower HDL-c levels. The CVR values, calculated both with the REGICOR and FRESCO formulas, were higher in men than in women, and associations were found with advancing age in both sexes and with lower educational level in men. These results are consistent with other studies that reported lower educational level and lower CVR among women and that CVR factors differ between sexes [41,47,69,70].

Cognitive performance, which was explored through diverse neuropsychological tests, worsened in both sexes with increasing age and in the majority of tests with lower educational level. This relationship was also observed with depression in men and CNS treatments in women in some tests. In terms of variables related to CVR, worse figures of HTN, HDL-c, and BMI lowered the score of several tests only for women, whereas physical inactivity and DM affected several test scores for both sexes, but a greater number of tests in the case of women. Smokers and former smokers, both men and women, performed better in some of the tests, in contrast to other studies that reported smoking to be a risk factor for cognitive impairment [71]. A prospective study of a Spanish cohort in 1993 found a negative effect of smoking on cognitive performance [72]. Given the unexpected results of the association between tobacco use and improved cognitive performance, we analysed possible confounding factors and found that smokers and ex-smokers were younger and had a higher level of education, which would justify better performance as both variables are associated with the same.

In the present study, AF did not affect any test results, although current evidence associates AF with a higher risk of cognitive impairment [73].

The present analysis revealed that higher CVR is associated with poorer cognitive performance, which is in line with the published scientific evidence on CVR and cognitive decline [33–40,69,74–76].

Women with high CVR compared to those with low CVR as measured by the REGICOR risk function did not show differences. With the FRESCO function, higher CVR was associated with poorer performance in memory, visuoconstruction, attention and processing speed. Men with high CVR with the REGICOR underperformed in global cognition, verbal intelligence, attention and psychomotor speed. Similarly, when estimating CVR with the FRESCO function, differences

were found between subjects with high and low CVR in memory, verbal intelligence, attention and psychomotor speed. Men with moderate CVR performed worse in global cognition compared to those with low CVR.

An additional consideration is the potential for reverse causation, but epidemiological evidence supports a direct causal relationship between cardiovascular risk factors and cognitive impairment. Longitudinal cohorts have shown that increased cardiovascular risk in middle age is associated with accelerated cognitive decline and increased risk of dementia [77,78]. Mendelian randomisation studies reinforce this relationship. A genetic predisposition to hypertension and dyslipidaemia is linked to an increased risk of Alzheimer's disease, suggesting a causal effect [79–81]. However, some findings may reflect reverse causality. In preclinical stages of dementia, changes in blood pressure, weight and cholesterol are observed, indicating that incipient neurodegeneration may modify cardiovascular parameters prior to diagnosis [82,83].

The test outcomes on which CVR had the greatest impact were the Immediate memory tests and the TMTA. Although a comparison between the two formulas to calculate CVR is not the subject of this study, the differences in cognitive performance by CVR in women only were found when the FRESCO equation was used.

The existing cross-sectional studies using the REGICOR function also reported worse cognitive performance in subjects with higher CVR. The comparison of outcomes was hindered by the fact that some studies did not disaggregate them by sex, used different age ranges, or considered different adjustment variables. Nevertheless, in agreement with the findings of the present study, the domains most influenced by CVR were attention, executive function, psychomotor speed [34,37,76,84], and memory [34,37,68]. These results are also in line with the fact that processing speed is the strongest predictor of overall cognitive performance in batteries for cognitive assessment [77]. A study conducted in Spain using the Framingham REGICOR function also found that the higher the CVR, the worse the cognitive performance in terms of speed and visual-motor coordination [40]. Similar results are found in longitudinal studies using the different variables of the Framingham function [33,35,47,69,85–89].

To our knowledge, there are no cross-sectional or longitudinal studies using the FRESCO risk equation.

## Strengths and limitations

One strength of our study to highlight is that the evaluated cohort is based on the general population, which facilitates the generalization of results with greater external validity.

The studied population was relatively young and did not present cardiovascular events. Another strength of this study was relating CVR to cognitive performance at early ages. This would enable controlling modifiable CVR factors and therefore decrease the risk of cognitive impairment. In light of the present findings, future research should be performed assessing this relationship at even younger ages which would provide evidence on the influence of elevated CVR among these populations. Another strength was that the model was adjusted for multiple known confounders (CVR factors or not) and outcomes were analyzed separately by sex, which is of relevance as CVR factors and predisposition to cardiovascular and cerebral disease have been shown to differ between women and men. Therefore, these findings should be considered when conducting this type of research.

Finally, two different functions were employed for estimating CVR. This could help to determine the most effective tool to evaluate the risk of cognitive deterioration. The estimation of CVR is an essential tool frequently employed in primary care to guide clinical decisions for controlling CVR factors ultimately aimed at avoiding cardiovascular events. However, this instrument could also be incorporated to facilitate strategies for the prevention of cognitive deterioration.

Some limitations must also be noted. The influence of eating habits could not be analyzed because of the lack of relevant data. The cross-sectional design of the study allows exploring the current cognitive performance of the subjects by their CVR but causal links cannot be inferred from the results. The long-term follow-up of this cohort will ideally enable furnishing this information.

REGICOR and FRESCO tools calculate the absolute 10-year risk of experiencing a cardiovascular event, so they cannot be applied to individuals who have already suffered an event. It would be interesting to analyse cognitive performance

in patients who have already suffered a cardiovascular event in order to broaden our understanding of the impact of cardiovascular events on cognitive function, and we therefore include this as a proposal for future complementary lines of research.

Finally, we would like to note that the performance of multiple statistical comparisons may increase the likelihood of obtaining statistically significant results by chance alone, thereby raising the risk of Type I errors, which should be taken into account when interpreting the findings.

### Applicability in clinical practice and research

This study allowed finding an association between CVR and cognitive performance in the Spanish population, which is characterized by a high percentage of CVR factors, but with few cardiovascular events. Longitudinal studies are warranted to confirm this association, identifying which cognitive functions are affected earlier with increasing CVR and also evaluating the optimal tools to detect cognitive deterioration as early as possible. Examining whether interventions targeting modifiable CVR factors can reverse or improve cognitive performance or mitigate the risk of cognitive decline is also a matter of interest.

On the other hand, these findings highlight the importance of assessing the joint effect of different risk factors rather than that of individual factors, as the latter approach could underestimate or overlook various effects. Therefore, this should be taken into consideration when analyzing the effect of CVR on cognitive performance.

Based on our conclusions, and in particular on prospective studies supported by longitudinal designs, the use of these tools, which are already used in clinical practice, especially in primary care, could be extended beyond their traditional applications. Identifying associations between cardiovascular risk and cognitive performance would not only help guide decisions regarding the indication and intensity of cardiovascular risk treatment aimed at preventing cardiovascular events, but would also promote a more comprehensive view of the patient, as it considers cognitive aspects and aims to reduce the incidence of cognitive impairment and dementia and prevent their progression. In this way, these tools, FRESCO and REGICOR may contribute to a more personalized and preventive approach integrating both cardiovascular health and cognitive well-being.

## Conclusions

Higher CVR was associated with worse cognitive performance in the Spanish population, both men and women, aged 55–75 years without cardiovascular disease. The most affected domains were memory, attention and processing speed. Therefore, high CVR is not only a predictor of a higher risk of suffering cardiovascular disease but also a poorer cognitive performance in middle ages.

## Supporting information

**S1 Fig. Directed Acyclic Graph (DAG) created with DAGitty v3.1 to identify paths of confusion between exposure (cardiovascular risk) and outcome (cognitive performance).**
(PDF)

**S1 Table. Baseline characteristics and outcomes of cognitive performance test by de presence of cardiovascular events (stroke, ischemic peripheral arterial disease).**
(PDF)

**S2 Table. Baseline characteristics of the sample and cardiovascular risk.** Comparison between participants with the lowest score in the MMSE-37 and the rest.
(PDF)

**S3 Table. Baseline characteristics of the sample and cardiovascular risk.** Comparison between participants with the worst score in the Inmediate recall test and the rest.
(PDF)

**S4 Table. Baseline characteristics of the sample and cardiovascular risk.** Comparison between participants with the worst score in the Delayed Recall test and the rest.
(PDF)

**S5 Table. Baseline characteristics of the sample and cardiovascular risk.** Comparison between participants with the worst score in the Word Accentuation test and the rest.
(PDF)

**S6 Table. Baseline characteristics of the sample and cardiovascular risk.** Comparison between participants with the worst score in the Verbal Fluency test and the rest.
(PDF)

**S7 Table. Baseline characteristics of the sample and cardiovascular risk.** Comparison between participants with the worst score in the Clock drawing test and the rest.
(PDF)

**S8 Table. Baseline characteristics of the sample and cardiovascular risk.** Comparison between participants with the worst score in the TMTA-1 and the rest.
(PDF)

**S9 Table. Baseline characteristics of the sample and cardiovascular risk.** Comparison between participants with the worst score in the TMTA-2 and the rest.
(PDF)

**S10 Table. Baseline characteristics of the sample and cardiovascular risk.** Comparison between participants with the worst score in the TMTA-1 errors test and the rest.
(PDF)

**S11 Table. Baseline characteristics of the sample and cardiovascular risk.** Comparison between participants with the worst score in the TMTA-2 errors test and the rest.
(PDF)

**S12 Table. Effect of cardiovascular risk, measured with the REGICOR and FRESCO equations, on cognitive performance in women.**
(PDF)

**S13 Table. Effect of cardiovascular risk, measured with the REGICOR and FRESCO equations, on cognitive performance in men.**
(PDF)

## Acknowledgments

We would like to thank Isabel del Cura González as well as the rest of staff in the Primary Care Research Unit of Madrid for their contribution to the design of the project and their support throughout the entire project.

We are grateful to all professionals participating in the NEurological DIsorders in Central Spain (NEDICES2 study): Hospital 12 de Octubre (R Trincado and C. Martin-Arriscado), participating Primary Care Research Unit of Salamanca (JL.

Alberca- Herrero, CM. Becerro-Muñoz, L. Fernández-del Campo, S. González-Sánchez, M. Meigide-García, AR, Menor-Odriozola, S. Mora-Simón, MP. Muriel-Díaz, E. Rodríguez-Sánchez, JA. Romero-Furones, O. Tamayo-Morales, J. Unzueta-Arce); Healthcare centre Arévalo, Ávila (T. Sobrino-Arroyo, P. Marqués-Macías, A. Benito-Pérez, CT. Martín-García Sancho, M. Mulero-San José, L. López-Gay, M. Jiménez-Nieto, C. López-Enríquez, MA. Jiménez-Carabias, E. Donis-Mulero, R. Blanco Marqués; J. López-Hebrero); Healthcare centre Cantalejo, Segovia (JF.Gil-García, C. Sanz-Herrero, MT. Calvo-Navajo, AM. de Lucas-Herrero, A. García-Luengo, Rincón-Heras, EM. Alvárez- de- Castro, E. Compes-Ribes, MJ. Monjas-Heras); Healthcare centre Las Calesas, Madrid (D. Martín-Acicoya, N. Calvo-Arrabal), Healthcare centre Comillas, Madrid (R. Calvo-Muller, P. Romera-Gutiérrez, MAP. Hernando-García) and Healthcare centre Fuentelarreina, Madrid (ML. Asensio-Ruiz, MC. Díaz-Laso, MV. Díaz-Puente, R. Ruiz Morote-Aragón, JM. Vizcaíno Sánchez-Rodrigo).

NEDICES2-RISK group collaborators by regions: Salamanca: Luis García-Ortiz[1], Manuel Ángel Gómez Marcos[1], Olaya Tamayo-Morales[1], Susana González Sánchez[1], Sara Mora-Simón[1], Jaime Unzueta-Arce[1], Paz Muriel-Díaz[1], Ana Menor-Odriozola[1], Belén Mateos-Montero[1], Alfonso Romero-Furones[1], José Luis Alberca-Herrero[1], Lucas Fernández del Campo-Carranza[1], Lourdes de la Rosa-Gil[1], Mercedes Meigide-García[1], Elena de Dios Rodríguez[1]; Arévalo, Ávila: Pilar Marqués-Macías[2], Ana Benito-Pérez[2], Candelas Teresa Martín-García Sancho[2], Teodoro Moreno-Sobrino[2], Ignacio Conde-Carrillo[2], Rita Morales-Hernández[2], Margarita Jiménez-Nieto[2], Cristina López-Enríquez[2]; Cantalejo, Segovia: Teresa Calvo-Navajo[3], Eugenio Pablo García-de Santos[3], Martín Oswaldo Riofrío-Pastor[3], Marianny del Carmen Guzmán-Jumelles[3], Noelia de la Esperanza-Esteban[3], Cristina Peláez-Martín[3], Elvira Compes-Ribes[3], Madrid: Health Centre Las Calesas: Diego Martín-Acicoya[4]; Health Centre

Comillas: Raquel Calvo-Müller[5], Elia Arranz-Martín[5]; Health Centre Fuentelarreina: María Luisa Asensio-Ruiz[6], María Concepción Díaz-Laso[6], María Victoria Díaz-Puente[6], Rafael Ruiz Morote-Aragón[6], José María Vizcaíno Sánchez-Rodrigo[6].

Lead author: Eugenio Pablo García-de Santos

Email: eugeniopablog@gmail.com

1 Institute of Biomedical Research of Salamanca (IBSAL). Primary Health Care Research Unit, the Alamedilla Healthcare Center, Gerencia Asistencial Atención Primaria, Servicio de Salud de Castilla y León (SACyL), Salamanca, Spain.

2 Healthcare Centre Arévalo, Gerencia Asistencial Atención Primaria, Servicio de Salud de Castilla y León (SACyL), Arévalo, Spain

3 Healthcare Centre Cantalejo, Gerencia Asistencial Atención Primaria, Servicio de Salud de Castilla y León (SACyL), Segovia, Spain.

4 Healthcare Centre Las Calesas, Gerencia Asistencial Atención Primaria, Servicio Madrileño de Salud (SERMAS), Madrid, Spain

5 Healthcare Centre Comillas, Gerencia Asistencial Atención Primaria, Servicio Madrileño de Salud (SERMAS), Madrid, Spain

6 Healthcare Centre Fuentelarreina (SERMAS), Gerencia Asistencial Atención Primaria, Servicio Madrileño de Salud, Madrid,

## Author contributions

**Conceptualization:** María del Canto de Hoyos-Alonso, Teresa Sanz-Cuesta, Isabel del Cura-González.

**Data curation:** Ester Tapias-Merino, María del Canto de Hoyos-Alonso.

**Formal analysis:** Ester Tapias-Merino, María del Canto de Hoyos-Alonso, Javier Rubio-Serrano, Beatriz Arregui-Gallego, Ileana Gefaell-Larrondo, Isabel del Cura-González.

**Funding acquisition:** Ester Tapias-Merino.

**Investigation:** Ester Tapias-Merino.

**Methodology:** Ester Tapias-Merino, María del Canto de Hoyos-Alonso, Javier Rubio-Serrano, Beatriz Arregui-Gallego, Ileana Gefaell-Larrondo, Isabel del Cura-González.

**Project administration:** Ester Tapias-Merino.

**Resources:** Ester Tapias-Merino.

**Software:** Ester Tapias-Merino.

**Supervision:** Ester Tapias-Merino, María del Canto de Hoyos-Alonso, Isabel del Cura-González.

**Validation:** Ester Tapias-Merino.

**Visualization:** Ester Tapias-Merino, María del Canto de Hoyos-Alonso, Javier Rubio-Serrano, Beatriz Arregui-Gallego, Ileana Gefaell-Larrondo, Isabel del Cura-González.

**Writing – original draft:** Ester Tapias-Merino, María del Canto de Hoyos-Alonso.

**Writing – review & editing:** Ester Tapias-Merino, María del Canto de Hoyos-Alonso, Javier Rubio-Serrano, Beatriz Arregui-Gallego, Ileana Gefaell-Larrondo, Emiliano Rodríguez-Sánchez, Israel Contador, Teresa Sanz-Cuesta, Félix Bermejo-Pareja, Isabel del Cura-González.

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
