## [Decision Letter · Decision Letter 0]

16 Oct 2025

Dear Dr. Tapias-Merino,

Kind regards,

Santhi Silambanan, MD, DNB

Academic Editor

PLOS ONE

Journal Requirements:

3. One of the noted authors is a group or consortium NEDICES2-RISK Group. In addition to naming the author group, please list the individual authors and affiliations within this group in the acknowledgments section of your manuscript. Please also indicate clearly a lead author for this group along with a contact email address.

5. In the online submission form, you indicated that regarding data exchange, the Ethics Committee for Research of the Hospital Universitario 12 de Octubre approved this research without considering the option of data sharing. The data contains sensitive clinical information about the patient, so there are ethical and legal restrictions to sharing the data set. The data are part of the NEDICES2-RISK study and can be requested by contacting the Primary Health Care Research and Innovation Foundation (FIIBAP) in the Community of Madrid at the email address fiibap@salud.madrid.org for the request of data

Additional Editor Comments :

The authors shall respond to all the queries raised by the reviewers

Reviewers' comments:

Reviewer's Responses to Questions

**Comments to the Author**

1. Is the manuscript technically sound, and do the data support the conclusions?

Reviewer #1: No

Reviewer #2: Yes

2. Has the statistical analysis been performed appropriately and rigorously?

Reviewer #1: Yes

Reviewer #2: Yes

3. Have the authors made all data underlying the findings in their manuscript fully available?

Reviewer #1: Yes

Reviewer #2: No

4. Is the manuscript presented in an intelligible fashion and written in standard English?

Reviewer #1: Yes

Reviewer #2: Yes

Reviewer #1: First of all I should say Authors did really valuable work on comparing CVR and cognitive performance, I just give some recommendation to improve their work:

How authors follow potential confounders in their studies?

How authors address recall bias of cohort study? Because Data extract from cohort study.

Explain how the study size was arrived at?

Explain how missing data were addressed?

Indicate number of participants with missing data for each variable of interest.

Reviewer #2: The introduction is comprehensive and well-referenced, giving a clear overview of the global and national context of dementia and cardiovascular disease. It correctly emphasizes the importance of modifiable risk factors and the need to study these relationships in Mediterranean populations.

That said, the section could be more focused. Several paragraphs repeat well-known information on dementia trends and cardiovascular epidemiology. The rationale for using the REGICOR and FRESCO equations in relation to cognition appears late and could be highlighted earlier.

The authors convincingly justify the inclusion of sex-stratified analyses, given known biological and lifestyle differences. However, the introduction would benefit from a more explicit hypothesis and a brief conceptual framework linking cardiovascular pathology (e.g., microvascular damage, inflammation) to specific cognitive domains such as memory and processing speed.

Selection bias: Excluding participants with prior cardiovascular events or those taking lipid-lowering drugs (for FRESCO) may skew the sample toward healthier individuals, underestimating the true relationship between CVR and cognition.

Outcome measurement: Although the neuropsychological battery is broad, the brief nature of some tests (e.g., 6-item recall, short fluency task) may reduce sensitivity to subtle cognitive differences.

Statistical approach: Dichotomizing cognitive scores into quartiles (≤25th percentile vs. >25th) leads to a loss of information and power. Modeling continuous outcomes could have provided more nuanced results. Additionally, the use of multiple comparisons without correction raises the possibility of false-positive findings.

Residual confounding: Important variables such as diet, alcohol intake, sleep, and socioeconomic status were not considered, though all are known to influence both cardiovascular and cognitive health.

The discussion section appropriately interprets the main results within the context of existing literature. The authors correctly note that their findings align with prior studies showing that higher cardiovascular burden correlates with poorer cognitive function. They also highlight that memory and processing speed were the domains most affected—consistent with vascular contributions to cognitive decline.

Nevertheless, the discussion could be more critical in tone. Several points could be strengthened:

The possibility of reverse causation should be explicitly addressed.

Mechanistic explanations (e.g., cerebral microvascular damage, systemic inflammation, impaired perfusion) are only briefly mentioned and could be elaborated.

The observation that smokers and ex-smokers performed better in some tests is intriguing but counterintuitive; this finding deserves cautious interpretation and deeper exploration, as it may reflect cohort or survivor effects rather than a genuine protective association.

The authors could also discuss the implications of their findings for preventive strategies in primary care, particularly since REGICOR and FRESCO are tools already used in clinical settings.

what does this mean?. If published, this will include your full peer review and any attached files.). If published, this will include your full peer review and any attached files.

**Do you want your identity to be public for this peer review?** For information about this choice, including consent withdrawal, please see our For information about this choice, including consent withdrawal, please see our Privacy Policy .

Reviewer #1: No

Reviewer #2: No

While revising your submission, please upload your figure files to the Preflight Analysis and Conversion Engine (PACE) digital diagnostic tool, https://pacev2.apexcovantage.com/ . PACE helps ensure that figures meet PLOS requirements. To use PACE, you must first register as a user. Registration is free. Then, login and navigate to the UPLOAD tab, where you will find detailed instructions on how to use the tool. If you encounter any issues or have any questions when using PACE, please email PLOS at . PACE helps ensure that figures meet PLOS requirements. To use PACE, you must first register as a user. Registration is free. Then, login and navigate to the UPLOAD tab, where you will find detailed instructions on how to use the tool. If you encounter any issues or have any questions when using PACE, please email PLOS at figures@plos.org . Please note that Supporting Information files do not need this step.

---

## [Author Response · Author response to Decision Letter 1]

30 Nov 2025

We would like to sincerely thank the Editor and the Reviewers for their thorough evaluation of our manuscript and for the constructive and insightful comments provided. We greatly appreciate the time and expertise invested in reviewing our work. We have carefully considered each observation and have revised the manuscript accordingly. Below, we provide a point-by-point response detailing the changes made and clarifying the issues raised.

In addition, we have incorporated the other modifications previously communicated to the Editor in August, which we were asked to introduce once the reviewers´ feedback became available. These additional modifications stem from the fact that, during the review period, we continued to perform additional analyses of the dataset in preparation for future publications. During this work, we identified that some subjects had been incorrectly classified with respect to their cardiovascular risk and we were also able to include the data from 4 additional participants in the revised analysis. While these corrections does not alter the main findings or overall conclusions of the previously submitted manuscript, we consider it important to update the text to maintain the highest level of accuracy and scientific rigor.

Changes made in response to the reviewers´comments are highlighted in yellow, while the additional modifications we propose to include are marked in blue in the revised manuscript.

We hope that the revised versión meets the expectations of the Editor and the Reviewers and we remain at your disposal foro any further clarification or adjustments needed.

Reviewer #1: First of all I should say Authors did really valuable work on comparing CVR and cognitive performance, I just give some recommendation to improve their work:

How authors follow potential confounders in their studies?

Thank you very much for your appreciation. To address the possible influence of confounding variables, we developed a Directed Acyclic Graph (DAG) representing the causal relationships between the study variables. This diagram allowed us to identify potential confounding pathways between exposure (cardiovascular risk) and outcome (cognitive performance) and determine the variables that should be adjusted in statistical analyses to obtain unbiased estimates of the causal effect.

We attach the image that has been included in the supplements section of the manuscript: Figure S1. Directed Acyclic Graph (DAG) created with DAGitty v3.1 to identify paths of confusion between exposure (cardiovascular risk) and outcome (cognitive performance).

In the methodology section of the manuscript, we have included the following sentence: A Directed Acyclic Graph (DAG) was constructed using DAGitty v3.1 to identify possible paths of confusion between exposure (cardiovascular risk) and outcome (cognitive performance).

The final criteria for the inclusion of confounding variables were based on (1) the development of the DAG, (2) the consideration of clinical variables relevant to our study objective, and (3) the identification of those that showed statistically significant differences.

Some potential covariates were not included in the model because they form part of the calculation of the risk value itself in the REGICOR and FRESCO tools, which could generate collinearity, affecting the stability of the model and the interpretation of the estimates. REGICOR includes the following variables in the risk calculation: age, sex, smoking, systolic and diastolic blood pressure, total cholesterol, HDL cholesterol, and diabetes. FRESCO, meanwhile, incorporates age, sex, smoking, systolic blood pressure, total cholesterol, HDL cholesterol, diabetes, and lipid-lowering treatment.

We have included the following sentence in the manuscript to clarify the above comment: Some potential covariates were not included in the model, as they are integrated into the risk value calculation in the REGICOR and FRESCO tools, and their additional inclusion could induce collinearity.

With all this in mind, we adjusted the generalised linear models for the following confounding variables: educational level (no education or primary education vs. secondary or superior education), obesity (presence if BMI ≥30 or absence), AF (presence or absence), depression (presence or absence), physical exercise (active or inactive), and treatment modulating the CNS (anxiolytics, neuroleptics, antidepressants, and antiepileptic drugs).

How authors address recall bias of cohort study? Because Data extract from cohort study.

We appreciate the reviewer´s concern regarding potential recall bias in our cohort study. In this regard, we would like to clarify the following:

- Objective clinical measurements: blood pressure, weight and height were obtained through direct measurement by trained clinical personnel. Therefore, these variables were no subject to participant recall and were not affected by recall bias.

- Medical history and comorbidities: information on participants´ medical conditions was collected by the study physicians during clinical encounters and verified against medical records. These data were not self-reported by participants.

- Neuropsychological assessmentes: all neuropsychological tests were administered in person during structured interviews conducted by qualified staff. The results reflect direct performance-based measures rather than recalled information.

- Missing data: any missing clinical information was retrieved from participants´ electronic medical records, ensuring accuracy and continuity of data without relying on participant memory.

For this reasons, recall bias is unlikely to have influenced our findings, as key variables were either directly measured, clinically assessed or extracted from medical records rather than self-reported.

We thank the reviewer for their insightful comments. Their observations made us realize that the manuscript did not explicitly describe how the study data were obtained. We will therefore include the following two paragraphs in the revised versión clarifying the data Collection procedures.

Clinical data, including blood pressure, weight and height, were obtained through direct measurements performed by trained clinical staff during study visits. Information on participants´ medical history and comorbidities was collected by the study physicians and verified using medical records. When necessary, missing clinical information was completed through review of the electronic medical records.

These neuropsychological tests were administered in person during structured interviews conducted by qualified personnel.

Explain how the study size was arrived at?

The study population was drawn from the NEDICES-2 (Neurological Disorders in Central Spain) cohort study. Patient recruitment took place in the primary care setting within the Spanish National Health System in the regions of Ávila, Madrid, Salamanca, and Segovia, Spain. The population in the study NEDICES2 originated from the list

of users (social security card holders) of the participating doctors in the included healthcare centers, who were then selected following randomized sampling of patients � 55 years-old stratified by gender and age (5-year intervals).

A specific sub-cohort, NEDICES2-RISK, was extracted from the original cohort study, which included all subjects in the NEDICES-2 cohort aged 55–75 years without dementia who underwent neuropsychological assessment. The sample size was determined by the number of participants who met these inclusion criteria.

Explain how missing data were addressed?

Participants with incomplete information on key variables for calculating cardiovascular risk were excluded from the analysis (complete case analysis). The proportion of missing data was low (0.52 %), only 5 participants, so it is unlikely that this procedure introduced significant bias.

In the case of neuropsychological tests, it was not possible to complete the full battery of tests in all patients, so an analysis of each test was performed with the patients who had completed it, excluding the rest. The number of patients analysed in each test is specified in the supplements (S2-S11).

Indicate number of participants with missing data for each variable of interest.

Attached here are the patients included in each CVR analysis and those who were able to complete each neuropsychological test:

Risk functions Included Excluded from the calculation in one of the risk functions Excluded from the calculation in both

Analysed with REGICOR 795 68 (age 75) 94 with cardiovascular events

5 with incomplete data

Analysed with FRESCO 555 308 (with lipid-lowering treatment)

TEST

Includad NA

MMSE-37 962 0

Immediate Recall test 960 2

Delayed Recall test 960 2

Word Accentuation test 944 18

Verbal Fluency test 958 4

Clock Drawing test 760 202

TMTA-1 949 13

TMTA-2 948 14

TMTA-Errors 1 945 17

TMTA-Errors 2 943 19

Reviewer #2: The introduction is comprehensive and well-referenced, giving a clear overview of the global and national context of dementia and cardiovascular disease. It correctly emphasizes the importance of modifiable risk factors and the need to study these relationships in Mediterranean populations.

That said, the section could be more focused. Several paragraphs repeat well-known information on dementia trends and cardiovascular epidemiology. The rationale for using the REGICOR and FRESCO equations in relation to cognition appears late and could be highlighted earlier

We thank the reviewer for this helpful observation. In response, we have streamlined the introduction by removing repetitive sentences related to dementia trends and cardiovascular epidemiology, ensuring that only the most essential background information is retained.

The authors convincingly justify the inclusion of sex-stratified analyses, given known biological and lifestyle differences. However, the introduction would benefit from a more explicit hypothesis and a brief conceptual framework linking cardiovascular pathology (e.g., microvascular damage, inflammation) to specific cognitive domains such as memory and processing speed.

We appreciate the reviewer´s valuable comment regarding the need for a more explicit hypothesis with a brief conceptual framework linking cardiovascular pathology to specific cognitive domains, including memory and processing speed. We agree that the evidence supporting these mechanism warrants clearer emphasis. Accordingly, we have revised the introduction to articulate the study hypothesis more explicity and to incorporate a concise conceptual framework supported by the appropiate references. We trust that these additions strengthen the manuscript and address the reviewer´s recommendation.

The added paragraph was the following:

Although the pathophysiological mechanisms of dementia remain unclear, several studies in general population samples have shown that microvascular dysfunction (assessed through biomarkers, imaging of small vessel disease or arterial stiffness) is associated with poorer performance in specific domains such as memory and processing speed.

Selection bias: Excluding participants with prior cardiovascular events or those taking lipid-lowering drugs (for FRESCO) may skew the sample toward healthier individuals, underestimating the true relationship between CVR and cognition.

Thank you very much for your contribution, which we consider very relevant and interesting. With regard to possible selection bias, participants with previous cardiovascular events were excluded, as the aim of our study was to assess how cardiovascular risk, estimated using the REGICOR and FRESCO tools, influences cognitive performance. These tools calculate the absolute 10-year risk of a cardiovascular event, so they cannot be applied to individuals who have already suffered an event.

Additionally, the FRESCO tool specifically excludes patients who are taking lipid-lowering treatment, as these drugs alter lipid levels and could skew the cardiovascular risk estimate. This exclusion is part of the risk calculator's own criteria.

However, we also find it interesting to analyse cognitive performance in patients who have already suffered a cardiovascular event in order to broaden our understanding of the impact of cardiovascular events on cognitive function, and we have therefore included this as a proposal for future complementary lines of research.

REGICOR and FRESCO tools calculate the absolute 10-year risk of experiencing a cardiovascular event, so they cannot be applied to individuals who have already suffered an event. It would be interesting to analyse cognitive performance in patients who have already suffered a cardiovascular event in order to broaden our understanding of the impact of cardiovascular events on cognitive function, and we therefore include this as a proposal for future complementary lines of research.

Outcome measurement: Although the neuropsychological battery is broad, the brief nature of some tests (e.g., 6-item recall, short fluency task) may reduce sensitivity to subtle cognitive differences.

We appreciate the reviewer´s observation regarding the potential reduction in sensitivity when using brief cognitive tests. We agree that shorter assessments may have limitations compared with more comprehensive neuropsuchological batteries.

Despite the low sensitivity of some of these tests, they have been added to the battery to provide a more comprehensive view of the different cognitive aspects. We use a total of seven different tests that take into account six different cognitive domains (global cognition, memory, premorbid intelligence, verbal fluency, visuoconstruction, attention, and psychomotor skills). Furthermore, we believe it is interesting to observe the performance of the brief tests because they are the most suitable for use in clinical practice.

Statistical approach: Dichotomizing cognitive scores into quartiles (≤25th percentile vs. >25th) leads to a loss of information and power. Modeling continuous outcomes could have provided more nuanced results. Additionally, the use of multiple comparisons without correction raises the possibility of false-positive findings.

We appreciate the reviewer´s comment and fully agree that, as noted, dichotomizing continuous outcomes can lead to a loss of information and statistical power. Nonetheless, we selected this approach to facilitate clinical interpretability and consistency with prior studies in the field.

The decision to divide cognitive scores into quartiles (≤25th percentile vs. >25th) was made because, at the start of the study, participants had very similar cognitive scores, resulting in a non-normal distribution. Attached is a table summarising the results of the normality tests, as well as the corresponding histograms for each neuropsychological test.

Below is the table with the p-values obtained after applying the Anderson-Darling test. When the p-value is extremely small for the Anderson-Darling test, R shows it in scientific notation as 3.7e-24.

Neuropsichological tests Anderson-Darling Test Kolmogorov-Smirvov Test

MMSE-37 3.7e-24 1.69e-21

Immediate Recall test 3.7e-24 8.76e-37

Delayed Recall test 3.7e-24 2.25e-36

Word Accentuation test 3.7e-24 1.55e-17

Verbal Fluency test 5.71e-13 7.23e-06

Clock Drawing test 3.7e-24 6.59e-107

TMTA-1 3.7e-24 8.75e-14

TMTA-2 3.7e-24 2.00e-22

TMTA-Errors 1 3.7e-24 1.81e-118

TMTA-Errors 2 3.7e-24 4.33e-74

All tests obtained a p-value lower than 0.05, therefore, the null hypothesis of normality is rejected, concluding that the data do not follow a normal distribution.

Quartile stratification allows for more accurate identification of participants with lower cognitive performance compared to those with higher performance, facilitating comparisons between groups and offering results that are more easily interpretable from a clinical point of view. Conversely, when modelling data continuously, this low variability between values could hinder the detection of significant associations and complicate the interpretation of results.

Thank you for your comment. We have included the following explanation of the reason for the quartile division in the methodology section.

The decision to divide cognitive score

---

## [Decision Letter · Decision Letter 1]

26 Jan 2026

Dear Dr. Tapias-Merino,

Thank you for submitting your manuscript to PLOS ONE. After careful consideration, we feel that it has merit but does not fully meet PLOS ONE’s publication criteria as it currently stands. Therefore, we invite you to submit a revised version of the manuscript that addresses the points raised during the review process.

We look forward to receiving your revised manuscript.

Kind regards,

Santhi Silambanan, MD, DNB

Academic Editor

PLOS One

Journal Requirements:

Additional Editor Comments:

Authors need to respond to the queries raised by reviewer 3

Reviewers' comments:

Reviewer's Responses to Questions

**Comments to the Author**

Reviewer #3: (No Response)

2. Is the manuscript technically sound, and do the data support the conclusions?

Reviewer #3: Yes

3. Has the statistical analysis been performed appropriately and rigorously?

Reviewer #3: Yes

4. Have the authors made all data underlying the findings in their manuscript fully available?

Reviewer #3: Yes

5. Is the manuscript presented in an intelligible fashion and written in standard English?

Reviewer #3: Yes

Reviewer #3: The authors have provided very detailed and comprehensive responses to previous reviewer comments. The changes they made to the manuscript in response to these comments have substantially improved the quality of the work.

The additional modifications of the analysis that the authors made beyond what was requested by the reviewers (highlighted with blue in the text) are well justified and indicates a critical and precise approach to their own work, which is commendable.

The study overall is well justified, the analysis is done carefully, and the interpretation of the results is mostly balanced.

I only have a few fairly minor comments:

1. While the authors provide a clear flow chart for how the analytical sample size was arrived at (Figure 1), they should describe this process much clearer in the text. For example, it is not clear that if the initial sample of 962 participants included only people between the ages of 55-74 years, why did they have to exclude a further 68 individuals who were 75 years or older when the REGIDOR score was analysed. In lines 138-141 on page 6, they should also mention the final sample sizes for the two scores. Particularly for the FRESCO score, a very large proportion of participants were excluded due to taking lipid lowering agents, and this needs to be explicitly presented in the text.

I`m also wondering whether the differences that we see between the REGIDOR and FRESCO score results are due to these differences in the analytical sample sizes, or due to real differences in the strength of the associations regarding these two scores.

2. I would also suggest that the main findings as described in lines 425-427 (page 24) should be much more nuanced. It is true that the results indicate significant inverse associations for some of the cognitive function indicators, but there were several non-significant associations as well. Therefore, the blanket statement of both risk scores being associated with poorer cognitive performance is not correct.

3. The authors did not really respond to the comment regarding multiple testing. As previous reviewer 2 correctly mentions, when so many associations are examined simultaneously, the issue of multiple testing and possibility of false positive associations may occur. As a minimum, I suggest mentioning this in the Limitations section of the manuscript, or maybe considering the application of Bonferroni correction or similar approach.

what does this mean?. If published, this will include your full peer review and any attached files.). If published, this will include your full peer review and any attached files.

**Do you want your identity to be public for this peer review?** For information about this choice, including consent withdrawal, please see our For information about this choice, including consent withdrawal, please see our Privacy Policy .

Reviewer #3: No

---

## [Author Response · Author response to Decision Letter 2]

24 Feb 2026

Dear Reviewer,

We would like to sincerely thank you for your careful review of our revised manuscript and for your very positive and constructive comments.

We greatly appreciate your recognition of the improvements made to the manuscript in response to the reviewers’ suggestions, as well as your acknowledgment of the additional analyses and clarifications we included beyond what was initially requested. Your assessment that these modifications strengthen the quality, rigor, and balance of the study is particularly encouraging.

We have carefully considered your comments and suggestions. Below, we provide our responses to the specific points you raised and indicate the corresponding revisions made in the manuscript.

Thank you again for your insightful comments and for helping us improve the quality of our study.

Kind regards,

Ester Tapias

---

## [Editor Report · Decision Letter 2]

2 Mar 2026

Cardiovascular risk and cognitive performance: a population-based cross-sectional study (NEDICES2-RISK).

PONE-D-24-29867R2

Dear Dr. Tapias-Merino,

We’re pleased to inform you that your manuscript has been judged scientifically suitable for publication and will be formally accepted for publication once it meets all outstanding technical requirements.

Kind regards,

Santhi Silambanan, MD, DNB

Academic Editor

PLOS One

---

## [Editor Report · Acceptance letter]

PONE-D-24-29867R2

PLOS One

Dear Dr. Tapias-Merino,

I'm pleased to inform you that your manuscript has been deemed suitable for publication in PLOS One. Congratulations! Your manuscript is now being handed over to our production team.

Kind regards,

on behalf of

Dr. Santhi Silambanan

Academic Editor

PLOS One